

# Entanglement islands in higher dimensions

**Ahmed Almheiri[1], Raghu Mahajan[1,2] and Jorge E. Santos[1,3⋆]**

**1** Institute for Advanced Study, Princeton, NJ 08540, USA
**2** Jadwin Hall, Princeton University, Princeton, NJ 08540, USA
**3** Department of Applied Mathematics and Theoretical Physics,
University of Cambridge, Wilberforce Road, Cambridge, CB3 0WA, UK

⋆ jss55@cam.ac.uk

## Abstract

It has been suggested in recent work that the Page curve of Hawking radiation can be recovered using computations in semi-classical gravity provided one allows for "islands" in the gravity region of quantum systems coupled to gravity. The explicit computations so far have been restricted to black holes in two-dimensional Jackiw-Teitelboim gravity. In this note, we numerically construct a five-dimensional asymptotically AdS geometry whose boundary realizes a four-dimensional Hartle-Hawking state on an eternal AdS black hole in equilibrium with a bath. We also numerically find two types of extremal surfaces: ones that correspond to having or not having an island. The version of the information paradox involving the eternal black hole exists in this setup, and it is avoided by the presence of islands. Thus, recent computations exhibiting islands in two-dimensional gravity generalize to higher dimensions as well.

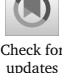

# 1   Introduction

The RT/HRT/EW formula [1–3] for computing entanglement entropies is a remarkable entry in the holographic dictionary. We are instructed to find a codimension-two surface in the bulk that minimizes the generalized entropy functional.[1] This codimension-two surface is called the quantum extremal surface (QES) and the value of the generalized entropy functional on the QES gives the entanglement entropy. Furthermore, the bulk region between the QES and the boundary, the entanglement wedge, can be reconstructed just using the knowledge of the corresponding boundary subregion [6–11].

The papers [12,13] considered the coupling of a large AdS black hole to a flat space bath region, allowing the black hole to evaporate. The entanglement entropy of the black hole was seen to undergo a first order phase transition following the appearance of a new nontrivial quantum extremal surface at late times.

Following this idea, [14] considered a two-dimensional gravity+matter theory, where the matter sector has a three-dimensional holographic dual. The main result of [14] is that the entanglement wedge of Hawking radiation at late times contains an "island" that lies in the interior of the black hole. This was also suggested in [12]. From a 2d viewpoint, this island is completely disconnected and spacelike separated from the naive domain of dependence of the region where the Hawking radiation lives. The 3d geometry connects these two pieces of the entanglement wedge.

The general lesson is that one should include contributions from islands in order to compute entanglement wedges and entropies of quantum systems coupled to gravity. The role of islands becomes crucial if there is a lot of entanglement between the bulk fields in the naive region and the island, for then, it can be beneficial to pay a cost proportional to the area of the island while incurring lots of savings in the bulk entropy. A prototypical case is to compute the entanglement entropy of the Hawking radiation that lies in the asymptotically-flat, weak-gravity region.

The state considered in [14] was time-dependent since the black hole is evaporating. In [15], the situation was simplified and it was demonstrated that islands exist even in large AdS black holes that are in equilibrium with a flat space bath region (and hence the geometry is static). All explicit computations in [15] were also for a two-dimensional gravity+matter theory, since this allows for some simple analytic expressions.

The goal of this note is to demonstrate that islands also exist in higher dimensions. For that purpose, we consider the equilibrium setup of [15], but in *four*-dimensional gravity+matter theories. To facilitate the computation of quantum extremal surfaces, we use the trick from [14] of taking the matter $CFT_4$ to have a five-dimensional holographic dual. In other words, we consider a Randall-Sundrum type setup with a 4d brane in a 5d ambient spacetime [16–18]. In this setup, quantum extremal surfaces in 4d become ordinary RT surfaces in 5d, and thus it becomes a tractable problem to compute them.

In particular, we will focus on the version of the information paradox described in section 4 of [15], see also [19].[2] This involves the thermofield double of a black hole coupled to, and in equilibrium with, a bath at some temperature. That is, there are two black holes, both coupled to their own baths.[3] One starts with a Cauchy slice through the middle of the Penrose diagram, and moves it forward in time on both sides. See figure 4. The question is what is the entanglement entropy of the union of the two baths as a function of this time? Naively,

---

[1]The generalized entropy functional [4,5] depends on a codimension-two surface and equals the area of this surface, plus the entropy of matter fields on the outer half of a Cauchy slice passing through this surface.

[2]This is different than the paradox discussed in [20]. See also [21] for a discussion of local operators behind the horizon in the eternal black hole.

[3]Note that unlike [14], the paper [15] did not assume that the matter sector has a three-dimensional dual. In other words the computations of [15] were all done in two dimensions.

this entropy increases linearly in time, forever. This happens because the bath is exchanging particles with the black hole: Hawking particles enter the bath and their entangled partners fall into the black hole. The mass of the black hole is not changing because we are in the Hartle-Hawking state, but the underlying exchange of quanta goes on. This is the analog of Hawking's calculation.

At late times, however, the entanglement wedge of the union of the bath regions contains an island that extends *outside* the horizon [15]. The generalized entropy of this QES saturates at late time, and is approximately equal to twice the Bekenstein-Hawking of a single black hole. This happens because the island contains the Hawking partners, and thus by including the island we save on the $S_{\text{bulk}}$ term in the generalized entropy. Thus, overall, the entropy grows linearly in time for a while before saturating.

In this note, we demonstrate that the same resolution works even in higher dimensions. The problem of setting up the above paradox in the 4d eternal black hole with a matter sector that has a 5d holographic dual reduces to finding a static 5d geometry with the correct boundary conditions. We construct this geometry numerically using the DeTurck trick [22–24]. This involves solving coupled PDEs for five functions of two variables each, see (17) for the ansatz for the line element. For a picture of the integration domain and the behavior of the space near the conformal boundaries and the Planck brane, see figure 2.

As already noted, quantum extremal surfaces in 4d become ordinary extremal surfaces in 5d. In the numerically constructed 5d geometry, we numerically find the extremal surfaces that are relevant for computing the entropy of the union of the two baths. There are two qualitatively different types of extremal surfaces, see figure 5. The extremal surface that dominates at early times goes through the horizon, and the entropy computed using this surface increases linearly in time. This is because of the stretching of space inside the horizon, as described in [25]. However, there is another extremal surface that dominates at late times. This extremal surfaces always stays outside the horizon and ends on the Planck brane. The entropy computed using this surface saturates at late times, essentially because, being completely outside the horizon, it does not get affected by the stretching of space inside the horizon.

Thus, our results provide a highly nontrivial check that the results of [12–15] are unchanged upon increasing the spacetime dimension: The information paradox is averted by the emergence of an island in the relevant entanglement wedge at late times.

Increasing the dimensionality of the setup of section 4 of [15] is a significant step forward because a possible criticism of [13–15] is that the explicit computations were only done in 2d AdS-JT gravity [26–29], which is known to be dual to an ensemble of Hamiltonians, rather than a single fixed Hamiltonian [30,31]. So one might wonder if 2d AdS-JT gravity is somehow not representative of a typical gravity theory, and that islands might not exist in higher dimensions. Even from the perspective of the gravity equations of motion, it is not completely obvious that the gravity computations of [13–15] generalize to higher dimensions. By our numerical construction, we explicitly provide this generalization.

The organization of this paper is as follows. In section 2, we discuss the action for the 5d gravity theory and the boundary conditions. In section 3, we describe the technique for numerically finding the static geometry. In section 4, we describe the relevant extremal surfaces, including the one that corresponds to having an island, and discuss how it avoids the information paradox in this setting. We conclude in section 5 with some discussion and future directions. Appendix A contains some details about the convergence of the numerical methods used.

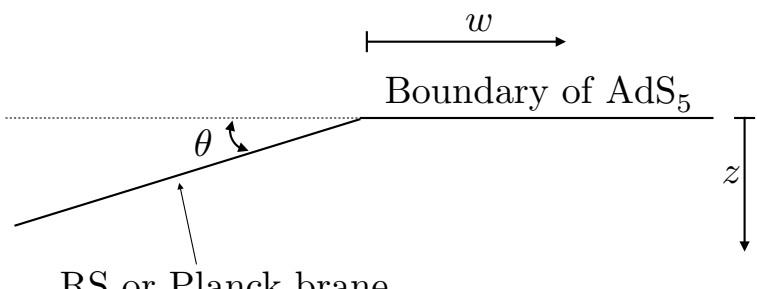

Figure 1: A simple geometry with a RS or a Planck brane, discussed in [32]. The RS or the Planck brane along lies along the locus $z = -w\tan\theta$. The induced geometry on the brane is AdS$_4$ with length scale (9). The angle $\theta$ is fixed by the tension parameter $\alpha$ in the action (1) via the relationship (6).

## 2 Setup of the problem

As mentioned in the introduction, following [14], we want to consider a "doubly-holographic" setup, but in higher dimensions. We take a 4d AdS gravity theory coupled to a matter CFT$_4$ that has a 5d holographic dual. We wish to consider a large black hole in this theory that is in equilibrium with a flat space bath region containing the same matter CFT$_4$. Thus, we are led to consider the following action:

$$I = \frac{1}{16\pi G_5}\int_{\mathcal{M}} d^5x\sqrt{-g}\left(R + \frac{12}{L^2}\right) + \frac{1}{8\pi G_5}\int_{\mathcal{B}} d^4x\sqrt{-h}\,(K - \alpha)\,. \tag{1}$$

Here $\mathcal{B}$ denotes the Planck or the RS brane [16] and it should be seen as one of the boundary components of the bulk spacetime. The quantity $L$ is the AdS$_5$ length scale, and $\alpha$ is proportional to the tension of the brane, see (8) below. The Gibbons-Hawking term at the UV boundary has been omitted to avoid clutter.

As discussed in detail in [14], the fundamental description of such a system should be taken to be a (2+1)-d holographic theory, coupled to a (3+1)-d bath system. In the first step, one replaces the (2+1)-d holographic theory with an AdS$_4$ spacetime. This AdS$_4$ spacetime is coupled to a flat space reservoir, and we have (3+1)-d matter fields propagating on this hybrid spacetime. In the second step, one assumes that the (3+1)-d matter fields are also holographic and replaces them with the (4+1)-d geometry. The (3+1)-d gravitational fields are represented by a Randall-Sundrum brane embedded inside this (4+1)-d geometry.

Varying the action (1) with respect to the metric gives us the Einstein equations

$$R_{AB} - \frac{R}{2}g_{AB} - \frac{6}{L^2}g_{AB} = 0 \qquad \text{on} \qquad \mathcal{M}\,. \tag{2}$$

Henceforth, upper case Latin indices will refer to the five-dimensional indices and lower case Latin indices will refer to coordinates along the brane. The boundary term in (1) allows for the usual Dirichlet boundary conditions, but we will infact impose the other possible alternative

$$K_{ab} - Kh_{ab} + \alpha h_{ab} = 0 \qquad \text{on} \qquad \mathcal{B}\,. \tag{3}$$

Recall from (1) that $h_{ab}$ is the induced metric on $\mathcal{B}$. This boundary condition is what one would call a Neumann boundary condition. In fact, imposing this Neumann boundary condition rather than the Dirichlet one is what allows a given boundary component of $\mathcal{M}$ to be called a Planck brane.

Before proceeding, let us review a simple Randall-Sundrum geometry [16,17], which has a Planck brane ending on the conformal boundary of AdS. This configuration was also considered in [32,33] in the context of AdS/BCFT. We start with pure AdS written in Poincaré coordinates

$$ds^2 = \frac{L^2}{z^2}\left(-dt^2 + dz^2 + dw^2 + dw_1^2 + dw_2^2\right).$$ (4)

Now consider the surface $z = -w\tan\theta$, where $\theta$ is some angle between 0 and $\pi/2$. We only keep the region $z > -w\tan\theta$ for $w < 0$. Of course, we always restrict to $z > 0$ even for $w > 0$. See figure 1.

We now want to implement the boundary condition (3), which is a form of the Israel junction conditions [34]. Computing the extrinsic curvature on the surface $w + z\tan\theta = 0$, we get

$$K_{ab} = \frac{\cos\theta}{L} h_{ab}.$$ (5)

Plugging this into (3), we get that the parameter $\alpha$ in the action (1) determines the angle $\theta$ via the relationship

$$\alpha = \frac{3\cos\theta}{L}.$$ (6)

From the Israel junction condition we know that the quantity

$$-\frac{1}{8\pi G_5}(K_{ab} - h_{ab}K) = \frac{1}{8\pi G_5}\frac{3\cos\theta}{L} h_{ab}$$ (7)

can be interpreted as the stress tensor of a codimension one object. This stress-tensor can be interpreted as arising from 3-brane with tension

$$T_3 = \frac{\alpha}{8\pi G_5}.$$ (8)

This value of the brane tension is consistent with the fact that the last term in the action (1) is equal to $\frac{\alpha}{8\pi G_5}$ times the worldvolume of the brane. Finally, note that by substituting $w = -z\cot\theta$ in the AdS$_5$ line element (4), and rescaling $z$, we see that the induced metric on the brane is nothing but AdS$_4$ with a length scale

$$L_4 = \frac{L}{\sin\theta}.$$ (9)

Let us now turn to the description of the actual numerical solution that we seek.

## 3 The numerical solution

### 3.1 The DeTurck trick

To find solutions, we will use the so-called DeTurck trick, which was first proposed in [22], and reviewed extensively in [23,24].

Let us first write the Einstein equation (2) in trace reversed form

$$R_{AB} + \frac{4}{L^2}g_{AB} = 0.$$ (10)

The idea is that, instead of directly solving (10), one considers the modified equation

$$R_{AB} + \frac{4}{L^2}g_{AB} - \nabla_{(A}\xi_{B)} = 0,$$ (11)

where $\xi^A := \left[\Gamma^A_{BC}(g) - \Gamma^A_{BC}(\bar{g})\right] g^{BC}$ is the so-called DeTurck vector, and $\bar{g}$ is a reference metric.[4] The reference metric $\bar{g}$ is required only to be regular and satisfy the same boundary conditions as $g$ on Dirichlet boundaries, but is otherwise arbitrary. In particular, if there are Neumann boundaries, the reference metric $\bar{g}$ is not required to satisfy the Neumann boundary condition there. The equation (11) is nice because the choice of gauge needed to solve Einstein's equations now appears as a choice of $\bar{g}$. Further, if we are looking for static solutions, then (11) together with either Dirichlet or Neumann boundary conditions is an elliptic problem, and is thus locally well-posed. (For Neumann boundaries, the DeTurck vector is also required to satisfy $\xi \cdot n = 0$, where $n$ is the normal vector to the boundary.) This is a major advantage over the original Einstein equation (10), whose character depends on the gauge choice even when seeking static solutions.

Solutions of (11) are not necessarily solutions of (10), because of the new added term $\nabla_{(A}\xi_{B)}$. Possible solutions with $\xi \neq 0$ are called DeTurck solitons. It can be shown that DeTurck solitons do *not* exist for static and certain stationary solutions of (11) with purely Dirichlet boundaries [35, 36]. In this case there is a complete equivalence between solutions of (11) and (10). On solutions with $\xi = 0$, the gauge choice is a generalisation of harmonic coordinates, given by $\triangle x^A = \Gamma^A_{BC}(\bar{g})g^{BC}$, where $\triangle$ stands for the scalar Laplacian in the metric $g$.

However, for Neumann boundary conditions on the metric, of the form (3), this has never been proved. Although in this case one cannot prove $\xi = 0$ on solutions of (11), one can still make progress because solutions of elliptic equations are locally unique. Hence, an Einstein solution cannot be arbitrarily close to a DeTurck soliton, and one should be able to distinguish the Einstein solutions of interest from DeTurck solitons by monitoring the quantity $\xi_A \xi^A$ appropriately.

## 3.2 The metric ansatz

Let us define $\widetilde{w} := w + z \cot\theta$. For numerical purposes, we take the domain of integration to be $\widetilde{w} > 0$ and impose (3) together with $\xi^A n_A = 0$ on the edge of the computation domain. Since we are interested in the Hartle-Hawking state, we want to have a bulk horizon that intersects the brane. Furthermore, the geometry should be such that at large $\widetilde{w}$ it should approach a five-dimensional planar black hole, whose line element reads

$$\mathrm{d}s^2_{\mathbb{P}} = \frac{L^2}{z^2}\left[-\left(1 - \frac{z^4}{z_+^4}\right)\mathrm{d}t^2 + \left(1 - \frac{z^4}{z_+^4}\right)^{-1}\mathrm{d}z^2 + \mathrm{d}\widetilde{w}^2 + \mathrm{d}w_1^2 + \mathrm{d}w_2^2\right]. \quad (12)$$

For numerical convenience we want to work with compact coordinates only, so we define a new coordinate $x \in (0, 1)$ via

$$\frac{x}{1-x} := \widetilde{w} = w + z \cot\theta. \quad (13)$$

Note that $x = 1$ is the asymptotic region $\widetilde{w} \to +\infty$. Finally, we change from $z$ to a coordinate $y$ where constant $t$ slices are manifestly regular at the event horizon $z = z_+$. One such choice is given by

$$y := \sqrt{1 - \frac{z}{z_+}}. \quad (14)$$

In terms of $(t, x, y, w_1, w_2)$ coordinates, the planar black hole reduces to

$$\mathrm{d}s^2_{\mathbb{P}} = \frac{L^2}{(1-y^2)^2}\left\{-y^2 G(y)\, y_+^2\, \mathrm{d}t^2 + \frac{4\,\mathrm{d}y^2}{G(y)} + y_+^2\left[\frac{\mathrm{d}x^2}{(1-x)^4} + \mathrm{d}w_1^2 + \mathrm{d}w_2^2\right]\right\}, \quad (15)$$

---

[4]Also, $\Gamma^A_{BC}(\mathfrak{g})$ is the Christoffel symbol associated with a metric $\mathfrak{g}$.

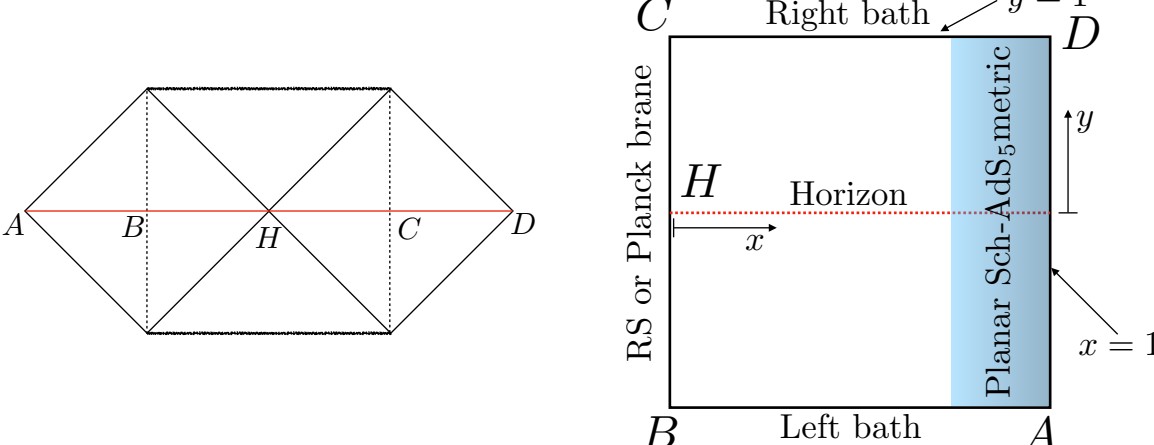

Figure 2: On the left, we show the Penrose diagram of the 4d geometry. We have a two-sided AdS black hole, with each side coupled to a bath. On the right, we show the integration domain used in the numerics $x \in (0,1)$ and $y \in (-1,1)$. The objective is to solve for five metric functions $Q_1, \ldots, Q_5$ of two variables each (17), in this domain. We numerically solve only in the region $y > 0$, the rest is obtained simply by symmetry. On the left edge of this diagram, at $x = 0$, we have the RS or the Planck brane where the 4d gravity region lives and the boundary condition (3) is imposed. On the top and bottom edges we have the two baths. As $x \to 1$, the metric approaches that of a 5d planar AdS-Schwarzschild black hole. The reader might find it useful to note the points $ABHCD$ on both diagrams. The precise induced geometry on the segment $BC$ is determined by the numerical solution, and the left picture is just a cartoon.

where

$$y_+ := z_+^{-1}, \quad \text{and} \quad G(y) := \left(2 - y^2\right)\left(2 - 2y^2 + y^4\right). \tag{16}$$

We are finally ready to present our metric ansatz:

$$\mathrm{d}s^2 = \frac{L^2}{(1-y^2)^2}\left\{ -y^2 G(y)\, y_+^2 Q_1\, \mathrm{d}t^2 + \frac{4Q_2\, \mathrm{d}y^2}{G(y)} + \right.$$
$$\left. \frac{Q_4}{(1-x)^4}\left[ y_+ \mathrm{d}x + 2(1-x)^2\, y\, Q_3\, \mathrm{d}y \right]^2 + y_+^2 Q_5\left(\mathrm{d}w_1^2 + \mathrm{d}w_2^2\right)\right\}. \tag{17}$$

Here $Q_I$, with $I \in \{1,2,3,4,5\}$, are functions of $(x,y) \in (0,1)^2$ to be determined by solving (11). For the reference metric we take the line element (17) with $Q_1 = Q_2 = Q_4 = Q_5 = 1$ and $Q_3 = \cot\theta$.

Let us now discuss the boundary conditions. At the horizon, located at $y = 0$, we impose Neumann boundary conditions for all variables, i.e. $\partial_y Q_I\big|_{y=0} = 0$ together with $Q_1(x,0) = Q_2(x,0)$, which in turn enforces the Hawking temperature to be

$$T_H = \frac{y_+}{\pi}. \tag{18}$$

At the conformal boundary, located at $y = 1$, and also at $x = 1$ we demand $g$ to approach the reference metric $\bar{g}$, that is to say $Q_1 = Q_2 = Q_4 = Q_5 = 1$ and $Q_3 = \cot\theta$. Finally, at the brane location, that is $x = 0$, we demand the boundary condition (3) together with $Q_3(0,y) = \cot\theta$ and $\xi^a n_a = -\xi^x = 0$. See figure 2 for a cartoon depiction of the integration domain.

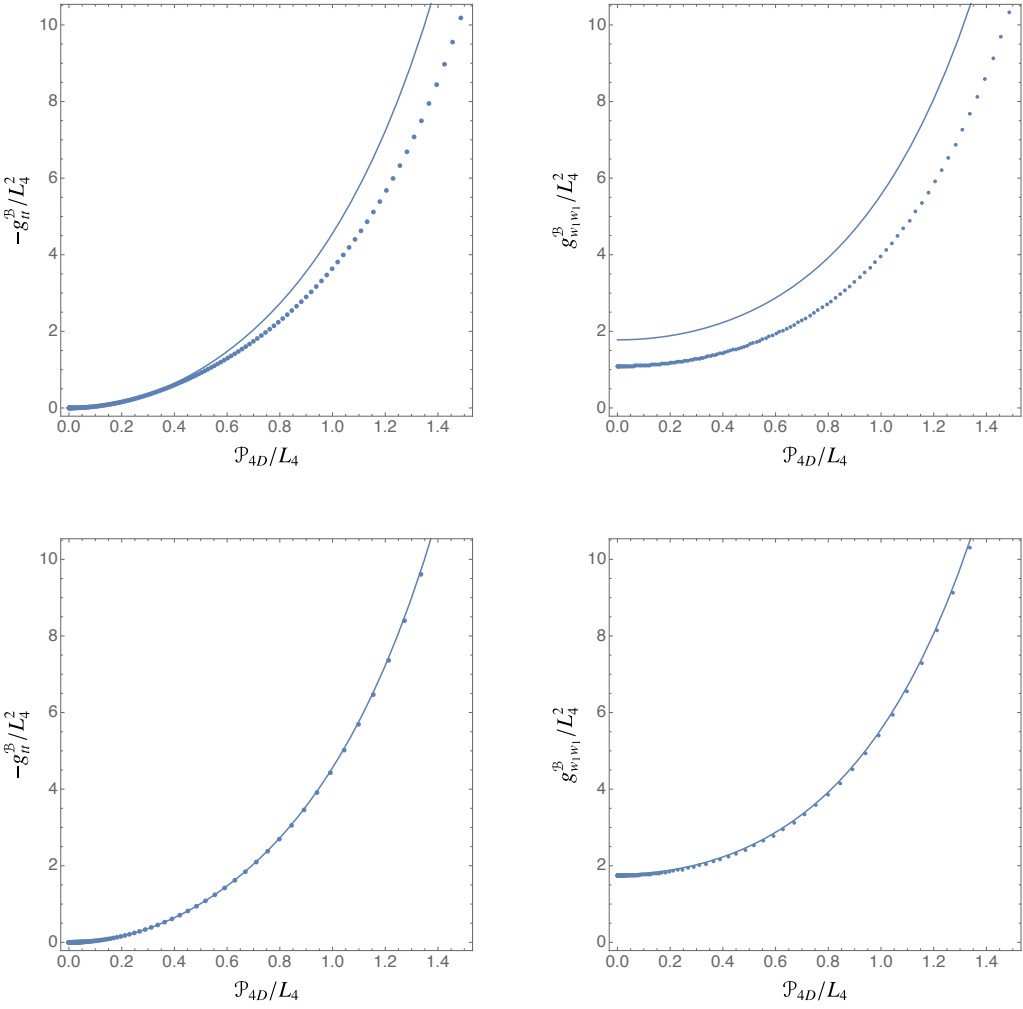

Figure 3: Plots of $-g_{tt}/L_4^2$ and $g_{w_1 w_1}/L_4^2$ on the RS brane (located at $x = 0$) as a function of the proper distance from the horizon $\mathcal{P}_{4D}$. In the top row, $\theta \approx 1.47113$, and in the bottom row, $\theta \approx 0.343024$. The blue disks correspond to the numerical data, and the solid blue lines are obtained from the 4d planar AdS black hole geometry. It is clear that as $\theta$ becomes smaller, the induced geometry on the brane gets closer to that of a 4d planar AdS black hole

These boundary conditions yield Robin-type boundary conditions on $Q_1$, $Q_2$, $Q_4$ and $Q_5$ at $x = 0$. It is then a simple exercise to show that (11) with such boundary conditions, gives rise to an elliptic problem [35].

To solve the resulting system partial differential equations, we used a standard pseudospectral collocation approximation on Chebyshev-Gauss-Lobatto points and solved the resulting non-linear algebraic equations using a damped Newton-Raphson method. The resulting method does *not* exhibit exponential convergence in the continuum limit due to the existence of non-analytic behaviour close the conformal boundary [37,38].[5] Instead, we will find a power law convergence as we approach the continuum limit.

### 3.3 Induced geometry on the brane

Recall that the boundary condition for the metric on the brane is Neumann rather than Dirichlet. Hence, the actual induced metric on the brane is determined numerically, and does not have a simple analytic expression. All we know is that there is a horizon at $y = 0$. In this subsection, we characterize the behavior of the induced geometry as $\theta$ becomes small. The upshot is that, in the limit $\theta \ll 1$, the induced geometry on the brane is close to that of a 4d planar AdS black hole.

In order to see this, consider the auxiliary line element of a four-dimensional planar black hole with horizon located at $Z_+$ and AdS$_4$ length scale $L_4$:

$$\mathrm{d}s_{4D}^2 = \frac{L_4^2}{Z^2}\left[-\left(1-\frac{Z^3}{Z_+^3}\right)\mathrm{d}t^2 + \left(1-\frac{Z^3}{Z_+^3}\right)^{-1}\mathrm{d}Z^2 + \mathrm{d}w_1^2 + \mathrm{d}w_2^2\right]. \tag{19}$$

Its associated Hawking temperature is given by $T_{4D} = \frac{3}{4\pi Z_+}$. If we want to match the temperature of our numerical solution reported in (18), we should impose $Z_+ = \frac{3}{4y_+}$.

To compare the line element (19) with the induced metric on the brane, we change to a new set of coordinates $\{t, \mathcal{P}_{4D}, w_1, w_2\}$, where $\mathcal{P}_{4D}$ is the proper distance from the horizon:

$$\frac{\mathcal{P}_{4D}(Z)}{L_4} = \int_Z^{Z_+} \frac{\mathrm{d}\tilde{Z}}{\tilde{Z}}\left(1-\frac{\tilde{Z}^3}{Z_+^3}\right)^{-\frac{1}{2}} = \frac{2}{3}\log\left(\sqrt{1-\frac{Z^3}{Z_+^3}}+1\right) - \log\left(\frac{Z}{Z_+}\right). \tag{20}$$

We then look at $-g_{tt}/L_4^2$ and $g_{w_1 w_1}/L_4^2$ as functions of $(\mathcal{P}_{4D})$, and compare with the results obtained from computing the same quantities using the induced metric on the brane.

More explicitly, the induced metric on the brane can be read off from (17) and is given by

$$\mathrm{d}s_{\mathcal{B}}^2 = \frac{L^2}{(1-y^2)^2}\left\{-y^2\,G(y)\,y_+^2\,Q_1(0,y)\,\mathrm{d}t^2 + 4\left[\frac{Q_2(0,y)}{G(y)} + y^2\cot^2\theta\,Q_4(0,y)\right]\mathrm{d}y^2\right.$$
$$\left. + y_+^2\,Q_5(0,y)\left(\mathrm{d}w_1^2 + \mathrm{d}w_2^2\right)\right\}. \tag{21}$$

Again, we can change to proper distance coordinates $\{t, \mathcal{P}_{4D}, w_1, w_2\}$ by defining

$$\frac{\mathcal{P}_{4D}(y)}{L_4} = \frac{2L}{L_4}\int_0^y \frac{\mathrm{d}\tilde{y}}{(1-\tilde{y}^2)}\sqrt{\frac{Q_2(0,\tilde{y})}{G(\tilde{y})} + \tilde{y}^2\cot^2\theta\,Q_4(0,\tilde{y})}, \tag{22}$$

where $L_4$ was is given by (9). Numerically, computing $\mathcal{P}_{4D}(y)$ can be tricky, because of the divergence of the integrand in the limit $\tilde{y} \to 1^-$. To bypass this difficulty, we consider instead

$$\frac{\mathcal{P}_{4D}(y)}{L_4} = \frac{2L}{L_4}\int_0^y \frac{\mathrm{d}\tilde{y}}{(1-\tilde{y}^2)}\left[\sqrt{\frac{Q_2(0,\tilde{y})}{G(\tilde{y})} + \tilde{y}^2\cot^2\theta\,Q_4(0,\tilde{y})} - \sqrt{1+\tilde{y}^2\cot^2\theta}\right]$$
$$+ \frac{2L}{L_4}\int_0^y \frac{\mathrm{d}\tilde{y}}{(1-\tilde{y}^2)}\sqrt{1+\tilde{y}^2\cot^2\theta}. \tag{23}$$

One can show that the integrand in the first line is finite[6] as $\tilde{y} \to 1^-$, while the integral in the second line can be readily done analytically, and carries all the divergences:

$$\frac{L}{L_4}\int_0^y \frac{\mathrm{d}\tilde{y}}{(1-\tilde{y}^2)}\sqrt{1+\tilde{y}^2\cot^2\theta} = \mathrm{arctanh}\left(\frac{y}{\sin\theta\sqrt{1+y^2\cot^2\theta}}\right) - \cos\theta\,\mathrm{arcsinh}(y\cot\theta). \tag{24}$$

---

[5]This non-analytic behaviour is a consequence of the DeTurck trick.

[6]Explicitly, we have $\displaystyle\lim_{\tilde{y}\to 1^-} \frac{2L}{(1-\tilde{y}^2)}\left[\left(\frac{Q_2(0,\tilde{y})}{G(\tilde{y})} + \tilde{y}^2\cot^2\theta\,Q_4(0,\tilde{y})\right)^{\frac{1}{2}} - \left(1+\tilde{y}^2\cot^2\theta\right)^{\frac{1}{2}}\right] = -L\sin\theta.$

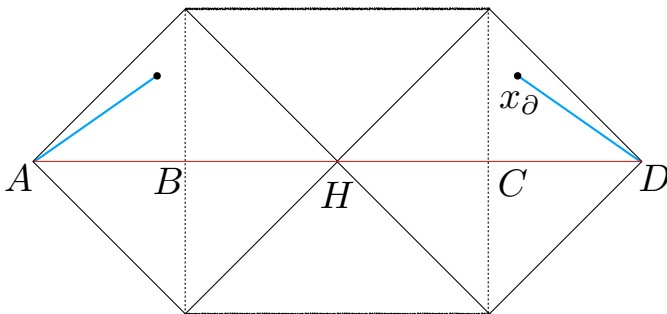

Figure 4: Shown here is a two-sided 4d black hole (with two of the spatial dimensions suppressed) coupled to two baths. See also figure 2. We want to compute the entanglement entropy of the union of the two blue regions shown. This diagram lives on the boundary of a static 5d spacetime whose exterior region was computed numerically in section 3.

We plot our comparisons in figure 3. The top row corresponds to $\theta \approx 1.47113$ and the bottom row corresponds to $\theta \approx 0.343024$. In all plots in figure 3, we have taken $y_+ = 1$. The blue disks correspond to the numerical data, and the solid blue lines are obtained from the 4d planar black hole geometry, as detailed above. The trend is clear: As $\theta$ becomes smaller, the induced geometry gets closer to that of a 4d planar AdS black hole.

# 4 Extremal surfaces and the island

Recall that we want to consider a version of the information paradox in 4d gravity theory coupled to a 4d matter sector. In this theory, we are considering a black hole coupled to, and in equilibrium with, a bath at nonzero temperature. We are also working in the two-sided purification, or the thermofield double, of the coupled system. So there are two black holes and two baths. We would like to compute the von Neumann entropy of the union of the left and the right bath regions as a function of time, where the time dependence is introduced by moving time forwards on both sides. See figure 4. The two-dimensional version of this problem was considered in section 4 of [15]. See also [19] and the recent paper [39].

We would like to compute 4d quantum extremal surfaces [3] for the union of the blue regions in figure 4. Since this is a very hard problem, we have made the simplification that the matter CFT$_4$ has a 5d holographic dual, as in [14], and so the 4d quantum extremal surfaces become ordinary 5d RT surfaces. Note that we are imagining toroidally compactifying the transverse directions to get IR-finite entropies.

## 4.1 Extremal surfaces at $t = 0$

We would like to compute these 5d RT surfaces [1]. More precisely, we would like to extract the extremal surfaces that anchor at the boundary at a given location $x = x_\partial > 0$. We will numerically compute extremal surfaces on the $t = 0$ slice of the line element (17).

As emphasized in [14], there are two extremal surfaces of interest emanating from $x_\partial$: the ones that penetrate the horizon, and the ones that end up anchoring on the brane (recall that the brane is located at $x = 0$), see figure 5. We will denote the area of the surface that penetrates the horizon by $\mathcal{A}_{\mathcal{H}}(x_\partial)$ and the area of the surface that ends on the Planck brane by $\mathcal{A}_{\partial\mathcal{M}}(x_\partial, y_{\mathcal{B}})$, where $y_{\mathcal{B}}$ is the value of $y$ at which this surface intersects the brane. Formally,

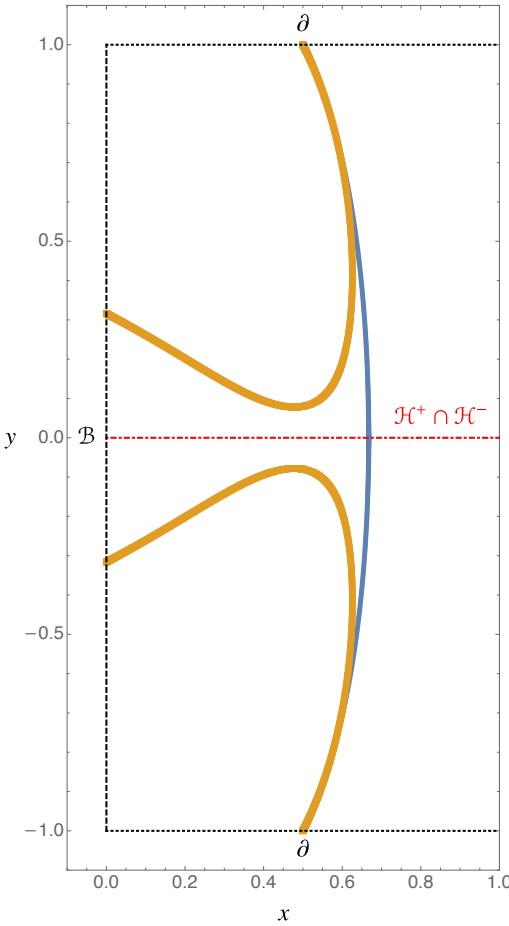

Figure 5: The two types of extremal surfaces, computed numerically at $t = 0$ in the background geometry found numerically in section 3. In this figure, we have taken $\theta = \pi/4$ and $x_\partial = 1/2$. The horizontal black dotted lines $\partial$ at the top and bottom are the left and right baths. The dashed black line $\mathcal{B}$ along the left edge is the location of the brane, which contains the 4d black hole. The horizontal red dashed-dotted line in the middle is the 5d bifurcate horizon, which meets the brane at the 4d horizon. Compare with figure 2. The orange curve corresponds to an extremal surface ending on the brane with $y_\mathcal{B} \approx 0.31602(1)$, while the blue curve correspond to an extremal surface that penetrates the bifurcating Killing surface smoothly. There is, in fact, a continuous family of orange extremal surfaces and there is a unique one amongst them with the smallest area, see figure 6.

both these areas are infinite, because of the divergence at the conformal boundary. However, the difference between these two is well defined.

Let us define the area difference

$$\Delta\mathcal{A}(x_\partial, y_\mathcal{B}) := \mathcal{A}_{\partial\mathcal{M}}(x_\partial, y_\mathcal{B}) - \mathcal{A}_{\mathcal{H}}(x_\partial), \tag{25}$$

which is finite for any pair $(x_\partial, y_\mathcal{B})$. We should also minimize this with respect to $y_\mathcal{B}$ and define

$$\Delta\mathcal{A}(x_\partial) := \min_{y_\mathcal{B} \in (0,1)} \Delta\mathcal{A}(x_\partial, y_\mathcal{B}). \tag{26}$$

We will simply compute $\Delta\mathcal{A}(x_\partial, y_\mathcal{B})$ for several values of $y_\mathcal{B}$ and look for a minimum. We will see that there is unique value of $y_\mathcal{B}$ that minimizes $\Delta\mathcal{A}(x_\partial, y_\mathcal{B})$.

Our extremal surfaces are parametrized by coordinates $\sigma^{\hat{\mu}}$, with $\hat{\mu} = 1, 2, 3$. For the surfaces that penetrate the horizon, we choose $\sigma^1 = y$, $\sigma^2 = w_1$ and $\sigma^3 = w_2$, so that the extremal surfaces can be parametrized by $x = F(y)$ in the $(x, y)$ plane. To compute such curves we look at the Euler-Lagrange equations derived from

$$
\begin{aligned}
S &= \int \mathrm{d}^3\sigma \sqrt{\det\left[(\partial_{\sigma^{\hat{\mu}}} x^{\dot{a}})(\partial_{\sigma^{\hat{\nu}}} x^{\dot{b}}) g_{\dot{a}\dot{b}}\right]} \\
&= \Delta w_1 \Delta w_2 \int_0^1 \mathrm{d}y \, \frac{L^3 y_+^2 Q_5(F(y), y)}{(1 - y^2)^3} \times \\
&\quad \left(\frac{4 Q_2(F(y), y)}{G} + \frac{Q_4(F(y), y)}{[1 - F(y)]^4}\left[y_+ \frac{\mathrm{d}F(y)}{\mathrm{d}y} + 2[1 - F(y)]^2 y Q_3(F(y), y)\right]^2\right)^{\frac{1}{2}},
\end{aligned}
\tag{27}
$$

where dotted indices run over the spatial coordinates $x, y, w_1, w_2$, but not over time. We will not present the explicit equations of motion following from (27) because they are not illuminating. Suffice it to say that they are second order ODEs in $F(y)$, and thus can only be solved once two boundary conditions are supplied. One of these boundary conditions is imposed at the conformal boundary, where we demand $F(1) = x_\partial$, while at the horizon we demand $F'(0) = 0$.

For the surfaces that end on the Planck brane, one has to proceed with more care, because if we try to think of these surfaces as a function $y(x)$ or $x(y)$, these functions will be multi-valued, see the orange curve in figure 5. To bypass this, we introduce two parametrizations in two different parts of the surface. For a range $y \in (1, y_c)$ we take $x = F(y)$, i.e. we choose $\sigma^1 = y$, $\sigma^2 = w_1$ and $\sigma^3 = w_2$. As boundary conditions we demand $F(1) = x_\partial$ and $F(y_c) = x_c > x_\partial$, which yields a unique solution in this interval for given values of $x_\partial$, $x_c$ and $y_c$. For $x \in (0, x_c)$ we choose $\sigma^1 = x$, $\sigma^2 = w_1$ and $\sigma^3 = w_2$ with $y = P(x)$. We view the resulting second order ordinary differential equation as an initial value problem, where we demand $P(x_c) = y_c$ and $P'(x_c) = F'(y_c)^{-1}$. Finally, we read off $y_{\mathcal{B}} = P(0)$ from the integration procedure.

For numerical stability, we found that it was crucial to use the same parametrization for both surfaces near the boundary, as the leading divergences in (25) were easier to cancel.

The results are shown in figure 5, where we plot an example of the two types of curves in the $(x, y)$ plane. In this figure we used $\theta = \pi/4$ and $x_\partial = 1/2$. Also, the value of $y_{\mathcal{B}} \approx 0.31602(1)$ for this specific plot. For the surface that ends on the brane, we vary $x_c$ and $y_c$, which in turn varies $y_{\mathcal{B}}$. As we do so, we compute $\Delta\mathcal{A}(x_\partial, y_{\mathcal{B}})$ as in the left panel of figure 6. In the right panel of figure 6, we zoom in close to the point where the horizon intersects the brane and find that $\Delta\mathcal{A}(x_\partial, y_{\mathcal{B}})$ is minimized for some value $y_{\mathcal{B}} = y_{\mathcal{B}}^\star > 0$. For the particular run shown in figure 6, we find that this occurs for $y_{\mathcal{B}}^\star \approx 0.067224(5)$. Since the minimum is very shallow, one might wonder whether this is a numerical artefact. To show that this is not the case, we also plot error bars in figure 6, which are estimated via the numerical convergence studies performed in appendix A.

## 4.2 Time dependence of the entropy and the island

As shown in figure 6, the surface that penetrates the horizon (blue in figure 5) has smaller area at $t = 0$ and thus is the correct RT surface to use. The time dependence we are considering involves moving the two sides forwards in time, see figure 4.

As in [15, 25], the area of of this surface increases (linearly after a few thermal times) as we perform the time evolution. As explained nicely in [25], the intuitive reason behind this is the stretching of space behind a black hole horizon [40, 41]. We would have an information paradox if this entropy increase continued forever, because the von Neumann entropy of

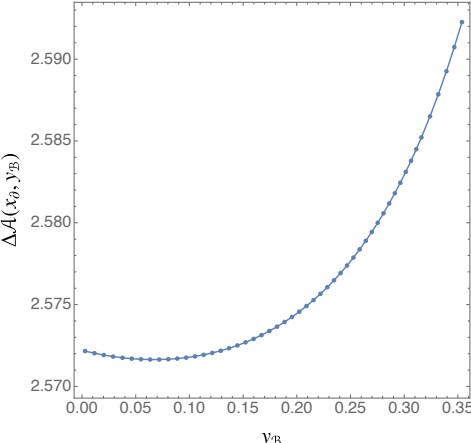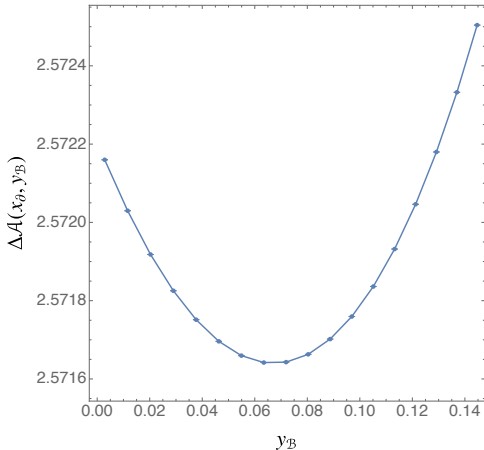

Figure 6: This figure depicts $\Delta\mathcal{A}(x_\partial, y_\mathcal{B})$ as a function of $y_\mathcal{B}$, computed for $x_\partial = 1/2$ and $\theta = \pi/4$. In the left panel we have $y_\mathcal{B} \in [0.0028(7), 0.35379(9)]$, whereas on the right we zoom in close to the point where the horizon intersects the brane. The surface corresponding to the minimum in this figure is the correct RT surface at late times.

the union of the two baths should saturate close to $2S_{\text{BH}}$. (Note that we are imagining the transverse directions to be toroidally compactified.)

The resolution is that the area of the surface that ends on the Planck brane (orange in figure 5) approaches approximately $2S_{\text{BH}}$ at late times, as in [15]. Again, the intuitive reason is that since this surface does not penetrate the horizon, it does not get affected by the stretching of space inside the horizon. Thus, the surface that ends on the Planck brane will win at late times.

The exchange of dominance between these two surfaces leads to an entropy that increases linearly for a while before saturating. This is the resolution of the information paradox in this setting.

Note that at late times, the entanglement wedge of the union of the left and the right baths contains an island. The island is the region on the left vertical line in figure 5 in between the two points where the orange curves intersect it.

## 4.3 Relation to Penington's work and comments on greybody factors

Higher-dimensional evaporating black holes were also discussed in section 2.4 of [12], where an argument using the intermediate value theorem was presented for the existence of a quantum extremal surface behind the horizon. The quantitative location of the QES in higher dimensions was left undetermined. While we believe that a more accurate computation will not qualitatively change the results in [12], at the same time it is desirable to obtain quantitatively the location of the QES.

Our work sidesteps the complications of an evaporating black hole and direct computations of $S_{\text{bulk}}$ by considering an eternal black hole in the doubly holographic setup. In this toy setup, we are able to make quantitative predictions because $S_{\text{bulk}}$ gets geometrized as the area of the five-dimensional RT surface.

We would also like to make some comments on greybody factors which are present in higher dimensions.[7] The direct computation of greybody factors is hard. In our doubly holographic setup, the effect of 4d greybody factors is packaged into the numerical 5d geometry.

---

[7]We thank the referees of SciPost for emphasizing the issue of greybody factors.

Thus, even though we have not been explicit about greybody factors, they are taken into account in our computations.

However, note that we have not explicitly computed the time dependence of the initial RT surface. In particular, we do not know the precise coefficient of the linear growth of entropy. It would be desirable to do so, but the time dependent calculation is numerically more involved and is beyond the scope of this work. In principle, one could compute that coefficient and compare it to the same coefficient in the Hartman-Maldacena setup [25], which does not have the RS brane. One could also compare the times at which the entropy saturates. We expect that the coefficient of linear growth in our setup with the RS brane should be smaller, and the time to saturation should be larger than the corresponding quantities in the Hartman-Maldacena setup because of greybody effects.

## 5 Discussion

Following section 4 of [15], we discussed a version of the information paradox in a four-dimensional black hole coupled to a bath in the Hartle-Hawking state. Time dependence is introduced by moving time forwards on both sides. We ask for the von Neumann entropy of the union of the left and right baths as a function of time, as depicted in figure 4. To facilitate computations of quantum extremal surfaces, the matter is described by a $CFT_4$ that has a five-dimensional holographic dual [14].

We numerically solved Einstein's equations using the DeTurck trick and found a static five-dimensional geometry having two flat UV boundaries and a Planck brane, see figure 2. This geometry has a bifurcate horizon that intersects the Planck brane. In this setup, quantum extremal surfaces in 4d become usual RT surfaces in 5d. We have computed the extremal surfaces that correspond to computing the entropy of the union of the left and the right baths. There are two types of surfaces, as shown in figure 5. One type of extremal surface (blue in figure 5) penetrates the horizon and is the dominant one at early times. However, its area increases as a function of time because of the stretching of space inside the horizon [25]. If there was no competing extremal surface, this would lead to an indefinite growth of entropy. However, we know that the entropy of the union of the two baths, being equal to the entropy of the two black holes, should saturate close to $2S_{\text{BH}}$. The resolution is that, in fact there is a second type of surface (orange in figure 5) that ends on the Planck brane. Its area saturates at $2S_{\text{BH}}$, and thus, it wins at late times. Overall, we get an entropy that grows linearly and then saturates.

This also means that the entanglement wedge of the union of the two baths contains an island at late times [15]. In figure 5, this island is the region on the left vertical edge between the two points where the orange curves intersect the vertical line.

The results of this paper unambiguously show that at least some of the gravity computations of [13–15] done in AdS-JT gravity generalize to higher dimensions. In particular, the microscopic fact that 2d AdS-JT gravity is dual to an ensemble of Hamiltonians, rather than a single one, plays no crucial role as far as gravity computations are concerned. One can speculate about the possibility that quantities computed using path integrals over metrics should always be interpreted as suitably-ensemble-averaged quantities, and that to reproduce *all* the features present in observables of normal unitarily evolving quantum systems, one perhaps needs stringy physics in the bulk and all sorts of additional effects. See [42] for a recent discussion of ensemble averages vs. unitary evolution in this context.

In conclusion, this paper provides the first setup where quantitatively precise entanglement islands [12–15] have been computed in higher dimensions. The conclusion is the same: Islands appear in entanglement wedge of the Hawking radiation at late times and this stops

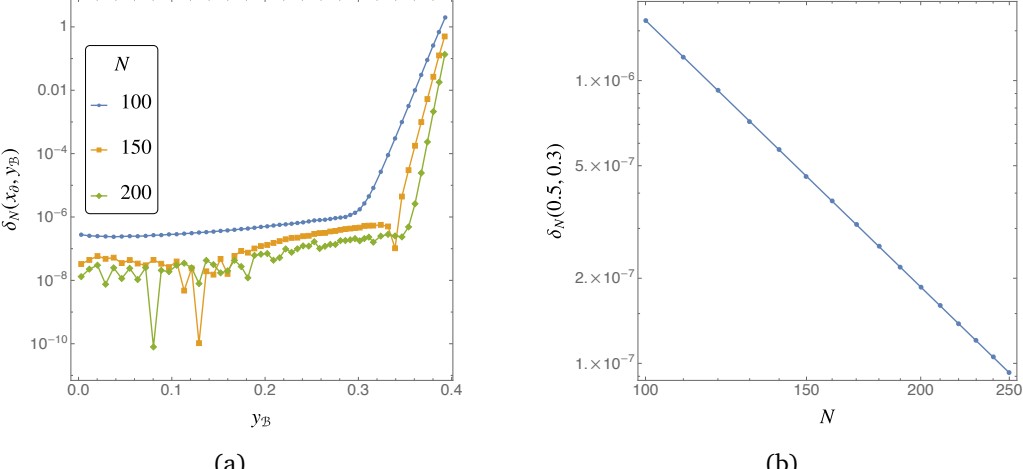

Figure 7: (a) Plot of $\delta_N(x_\partial, y_\mathcal{B})$ computed for several values of $N$ labeled in the plot. For $y_\mathcal{B} \in [0.0028(7), 0.35379(9)]$ the relative error is smaller than $10^{-6}$. (b) Plot of $\delta_N(0.5, 0.3)$ in a $\log-\log$ scale computed for several values of $N$. The numerical data is represented by the blue disks, and the solid blue line is a best fit curve which yields $\delta_N(0.5, 0.3) \sim N^{-3.13}$.

the indefinite growth of von Neumann entropy, giving an answer consistent with unitarity and a finite density of states.

There are quite a few natural extensions of our work. We found the static geometry and the two types of extremal surfaces numerically at $t = 0$, and then used general reasoning to deduce the time dependence of the areas. It would be interesting to explicitly compute the time dependence of the extremal area surfaces. It would also be interesting to see if one can make any analytic statements in the limit $\theta \to 0$. This is the limit where the length scale of AdS$_4$ (9) goes to infinity. Finally, it would be interesting to see if the scenario of "uberholography" [43], found to hold in the 2d/3d setup of [14] in the recent paper [44], persists in higher dimensions.

### Acknowledgments

We thank Juan Maldacena and Ying Zhao for many discussions and collaboration in the initial stages of this work. A. A. is supported by funds from the Ministry of Presidential Affairs, UAE. R. M. is supported by US Department of Energy grant No. DE-SC0016244. J. E. S. is supported in part by STFC grants PHY-1504541 and ST/P000681/1. J. E. S. also acknowledges support from a J. Robert Oppenheimer Visiting Professorship. This work used the DIRAC Shared Memory Processing system at the University of Cambridge, operated by the COSMOS Project at the Department of Applied Mathematics and Theoretical Physics on behalf of the STFC DiRAC HPC Facility (www.dirac.ac.uk). This equipment was funded by BIS National E- infrastructure capital grant ST/J005673/1, STFC capital grant ST/H008586/1, and STFC DiRAC Operations grant ST/K00333X/1. DiRAC is part of the National e-Infrastructure.

## A  Numerical convergence

In this appendix we study the numerical convergence of our numerical method. To discretize the PDEs we use a pseudo-spectral collocation scheme on two Chebyshev grids along the $x$ and $y$ directions. We then solve the resulting nonlinear algebraic equations using a standard damped Newton-Raphson algorithm. See [24] for a review of such methods applied in the

context of the Einstein equation.

Our main figure of merit for extremal surfaces is figure 6, so we shall use it to study numerical convergence. In figure 7a we plot

$$\delta_N(x_\partial, y_\mathcal{B}) := \left|1 - \frac{\Delta\mathcal{A}^N(x_\partial, y_\mathcal{B})}{\Delta\mathcal{A}^{N+50}(x_\partial, y_\mathcal{B})}\right|,\qquad(28)$$

as a function of $y_\mathcal{B}$. Here, $\Delta\mathcal{A}^N(x_\partial, y_\mathcal{B})$ stands for computing $\Delta\mathcal{A}(x_\partial, y_\mathcal{B})$ using two Chebyshev grids, along $x$ and $y$, each with $N$ gridpoints. The range quoted in the caption of figure 6 corresponds to a relative error of no larger than $10^{-6}$. This explicitly shows that the shallow minimum of figure 6 is clearly resolved for resolution with $N \geq 200$. All the plots in the main text were generated with $N = 250$.

To extract the convergence of the method, we fix $x_\partial = 1/2$ and $y_\mathcal{B} = 0.3$ and vary $N$. Fixing other values of $x_\partial$ or $y_\mathcal{B}$ give very similar results. The results are displayed in figure 7b, where the observed behaviour is consistent with power law convergence $\delta_N(0.5, 0.3) \sim N^{-3.13}$. This in turn agrees with the non-analytic behavior of the DeTurck gauge close to the conformal boundary that was uncovered in [37, 38].

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
