# Peer review of "Entanglement islands in higher dimensions"

_SciPost Physics, doi:SciPost Phys. 9, 001 (2020)_

## Round 2 · Referee Report · Anonymous (Referee 1) · 2020-2-12

Strengths

1- Clear, well-motivated, and interesting goal (find island in higher dimensional BH setup). 2- Good, logical explanation of method and results.

Weaknesses

1- Does not discuss the existing higher dimensional analysis in their reference 12, by Penington. 2- Unclear why the RS brane is allowed to have RT surface dynamically end on it and yet be considered part of the `boundary.’

Report

This is a thorough, clear analysis that answers an interesting question. The authors have done a service for the field by writing this up.

Two comments:
There could be some improvement in how these results are related to prior work. In particular, in reference [12], Penington argues for the existence of these islands in general dimensions. While he does make simplifying assumptions (spherical symmetry), his arguments are relevant to the goal of this paper and should be acknowledged.

How are we supposed to understand the seemingly-inconsistent treatment of the RS brane? Sometimes it acts like part of the "bulk’’ (because the RT surface is allowed to end on it at a dynamical location), and other times it acts like part of the "boundary’’ (because its degrees of freedom are understood to be entangled with the "true boundary’’ degrees of freedom). Is one of these interpretations more correct, e.g. that it should only be understood as part of the bulk (in which case how do we understand its entanglement with the "true boundary’’)? Or perhaps is the answer unknown, and you are simply demonstrating that treating the RS brane this way gives an interesting answer?

Requested changes

1- Add a sentence or two about Penington’s higher-dimensional island arguments. If what you are doing is more impressive for certain reasons, mention! 2- Add a couple sentences or a footnote about how we should think of the RS brane. As part of the bulk, part of the boundary, or somehow both?

  • validity: high
  • significance: high
  • originality: high
  • clarity: top
  • formatting: perfect
  • grammar: perfect

Author:  Raghu Mahajan  on 2020-05-27  [id 840]

(in reply to Report 1 on 2020-02-12)

Thank you very much for the report! We have addressed your requested changes as follows:

1- We have added a new section 4.3 comparing our work to Penington's

2- We have added some discussion about the fundamental microscopic description of the system under study, and its two gravitational avatars. This is on top of page 5.

---

## Round 2 · Referee Report · Anonymous (Referee 2) · 2020-4-6

Strengths

  1. Generalises the doubly holographic calculations showing the emergence of an "island" from [14] to higher dimensions. A useful and valuable goal.

  2. Technically precise and clearly presented.

Weaknesses

  1. Little discussion of the qualitative differences between two-dimensions and higher-dimensions

Report

The paper uses tools from numerical relativity to calculate the Page curve of an equilibrating black hole in a doubly holographic theory with 3+1 dimensions. It is a natural and important follow-up to the previous work on the subject (e.g. [12-15]), which mostly focussed on two-dimensional (or effectively two-dimensional) theories (although note the discussion of higher-dimensions and grey-body factors in [12]). Two-dimensional gravity theories are more analytically tractable but are obviously less similar to reality.

Since the main technical difference between two-dimensions and higher-dimensions is the existence of grey-body factors, it would be nice to have more discussion about how these effects can be seen in the doubly holographic description, where they are not particularly manifest. Ideally it would be nice to have a calculation where some parameters of the black hole can be dialed to change the grey-body factors, so that its effect on the growth of the doubly-holographic HRT surface can be seen. This may be impractical, of course. Regardless, some more discussion would be nice.

Requested changes

  1. It would be good to see an explicit comparison of the results in the present paper with the results about higher-dimensions and greybody factors from [12] (presumably the authors would note that in [12] the location of the QES was only calculated up to unknown O(beta) corrections in the retarded time -- it would be nice to see this explicitly commented on however).

  2. Some discussion how grey-body factors affect the equilibration process and how this can be seen in the doubly holographic description.

  • validity: high
  • significance: high
  • originality: high
  • clarity: top
  • formatting: perfect
  • grammar: perfect

Author:  Raghu Mahajan  on 2020-05-27  [id 841]

(in reply to Report 2 on 2020-04-06)

Thank you for the report. We have addressed the requested changes as follows:

We have added a new section 4.3 that includes a discussion of results of reference [12]. We have also added some comments on greybody factors.

---

## Round 3 · Referee Report · Anonymous · 2020-6-2

Report

The authors have addressed my previous points. I fully support publication.

---

## Round 3 · Referee Report · Anonymous · 2020-6-4

Report

The changes look good. I fully support publication.

---

## Round 3 · Author Response

We have added discussion about the fundamental microscopic description of the system under study. We have also added some comparison to Penington's work and comments on greybody factors.

---

## Round 3 · List of Changes

1) We have added discussion about the fundamental microscopic description of the system under study, and the two gravitational avatars. This discussion is on the top of page 5.

2) We have also added some comparison to Penington's work and comments on greybody factors. This is in the new section 4.3

---

## Editorial Decision

published